# *Juedi Tiantong*: The Religious Basis of the Relationship between *Tian* and Man in Ancient China

**Zhejia Tang and Xuedan Li \***

School of Philosophy, Zhejiang University, Hangzhou 310058, China; tangzhejia@hzcu.edu.cn
\* Correspondence: xuedanli@hzcu.edu.cn

**Abstract:** *Juedi Tiantong* occurred in ancient China and was the critical foundation for understanding the relationship between *Tian* and man in China. From the perspective of conceptual history, *Juedi Tiantong* not only shaped the metaphysical dimension of the concept of *Tian*, but also transformed the original religious form of communication between man and natural gods into the unity of human nature and *Tiandao*, which liberated the relationship between *Tian* and man from the religious field. Therefore, *Juedi Tiantong* should be regarded as the critical basis of the unity of heaven and man in Chinese philosophy. Furthermore, as an important religious revolution, *Juedi Tiantong* also affected people's understanding of nature, which was mainly reflected in the recognition of astronomy and calendar reform. In ancient China, it was difficult to distinguish between humanity and astronomy, science and religion, and rationality and divinity. In this case, *Juedi Tiantong* also abstracted *Tian*, originally representing the physical sky, into a metaphysical concept. Accordingly, the concept of *Tian* in Chinese philosophy has not developed the same meaning of nature as Western civilization.

**Keywords:** *Juedi Tiantong*; the relationship between *tian* and man; religion; nature; the unity of heaven and man

## 1. Introduction

*Juedi Tiantong* 絕地天通 (the severance between *Tian* and man) was a religion-based reform of great significance that happened in ancient China. The exact time of this religious reform is still subject to debate among historians, but most scholars agree that Zhuanxu 顓頊, the leader of *Juedi Tiantong*, lived during the period from approximately 3000 B.C. to 2500 B.C. (Z. Wang 2013, p. 381). The main reason for this revolution was likely due to the war that occurred during the integration of tribal states at that time (D. Yu 2005, pp. 11–18). This was a religious revolution, lasting hundreds of years, which changed the relationship between *Tian* 天 and man. *Juedi Tiantong* literally means the severance of communication between *Tian* and man. Before *Juedi Tiantong*, all shamans were able to invite and talk to the gods at any time. However, Zhuanxu, the martial and political leader with the highest prestige at that time, announced a religious policy that only himself and a few designated shamans were allowed to communicate with *Tian* and all other people were prohibited from sacrificing themselves to *Tian*.

As a major religious reform in ancient China, *Juedi Tiantong* has always been a hot topic among scholars. Chinese scholar Liang Tao 梁濤 once summarized that the academic articles discussing the influence of *Juedi Tiantong* can be broadly divided into three categories, namely, the theory of religious reform represented by Xu Xusheng 徐旭生, the theory on the monopoly of political rights represented by Zhang Guangzhi 張光直, and the theory on the specialization of *Wu* 巫 (shaman) represented by Chen Lai 陳來 (Liang 2022, pp. 47–54). The first category focuses on the religious issues and aims to explore the ideological changes brought about by *Juedi Tiantong*. Xu Xusheng 徐旭生 believes that Zhuanxu transformed primitive shamanism into a progressive religion, and the establishment of this religion system enabled the rapid development of Chinese culture (Xu 2023,

p. 31). Liu Wei 劉偉 believes that *Juedi Tiantong* was a religious event, rather than a simple historical event. This religious reform made the shamans specialized and the kingship sanctified, which facilitated the formation and maturity of the ancient state religion, and also preserved the core elements of the original shamanism (W. Liu 2021, pp. 125–32). The second research category focuses on the political issues and discusses the monopoly of political power demonstrated by *Juedi Tiantong*. Yang Xiangkui 楊向奎 believes that *Juedi Tiantong* meant that "the King monopolized the mediators between *Shangdi* 上帝 and human" (X. Yang 1962, p. 164), which implied the monopoly of political power through religious reform. On this basis, Zhang Guangzhi 張光直 further explored the construction of political authority and argued that "this mythological story provided clues to the central role of shamans in ancient Chinese politics. After *Juedi Tiantong*, only those who controlled the path to *Tian* had enough wisdom and political power to rule the state" (G. Zhang 2016, p. 36). The third research category focuses on the specialization of shamans in ancient China and discusses the transformation of shamanic culture. In *Ancient Religion and Ethics: The Root of Confucianism*, Chen Lai 陳來 regards shamanic culture as the source of Confucianism and regards *Juedi Tiantong* as an important link between shamanism and Confucianism (L. Chen 1996, p. 5). Chen believes that *Juedi Tiantong* resulted in the specialization of shamans, after which ritual specialists became the only group eligible to communicate with *Tian*. Li Xiaoguang 李小光 believes that Zhuanxu's reform realized a monopoly in terms of the path to *Tian*, but did not realize the unity of beliefs, which resulted in the establishment of polytheistic religion in China (X. Li 2008, pp. 138–41). Zhang Zhen 張震 and Su Huimi 蘇薈敏 argue that *Juedi Tiantong* was a starting point in the self-awareness of shamans and that this event was of great aesthetic significance that could not be ignored (Zhang and Su 2016, pp. 29–33+124).

Foreign scholars have also paid attention to the study of *Juedi Tiantong*. For instance, Yujin Lee believes that *Juedi Tiantong* can be understood from both historical and mythological perspectives, but the real value of the material concerns its implied historical authenticity (Lee 2002, pp. 445–71). Gilles Boileau believes that *Juedi Tiantong* brought order and differentiation to religious matters, in line with the late Zhou ideology of sacrifice and religion. *Wu*, as the head of a hierarchy of officers, appeared to be an elaboration of the relationship between religion and society, relevant within the framework of the ritual thinking of this period, particularly the ideology of sacrifice (Boileau 2002, pp. 350–78). Michael J. Puett also focused on the transition of shamans into ritual specialists caused by *Juedi Tiantong*; he argued that spirits and humans should be separated and placed within a proper hierarchy of functions and, after *Juedi Tiantong*, *Wu* were not shamans at all (Puett 2022, p. 107).

In recent years, more and more scholars have begun to pay attention to the relationship between *Tian* and man contained within *Juedi Tiantong*. Zhang Rulun 張汝倫 believes that *Juedi Tiantong* played a critical role in modeling the relationship between *Tian* and man (R. Zhang 2019, pp. 52–58). Zhao Guangming 趙廣明 believes that from *Juedi Tiantong* to *Zhili Zuoyue* 制禮作樂 (the establishment of a system of rites and music), the king was able to construct a rights-based social hierarchy by utilizing the relationship between *Tian* and man. This tradition was strengthened by Confucius, and the relationship between *Tian* and man was finally justified by Zhuangzi 庄子 (G. Zhao 2023, pp. 11–20). Xiao Qi 肖琦 believes that understanding *Juedi Tiantong* as the process of specialization of shamans or the centralization of imperial power does not correspond with its original meaning. *Juedi Tiantong* which is recorded in *Shangshu*, the *Lüxing* chapter 尚書·呂刑reflected that Chong 重 and Li's 黎 tribe defeated the tribe of Miao 苗, which is consistent with the Mandate of Heaven of Zhou, and is also not in conflict with the concept of the unity of heaven and man (Xiao 2019, pp. 37–43). As a ban on religion, *Juedi Tiantong* literally means cutting off communication between the people on earth and the gods in heaven, the consequence of which should be the creation of a pure, secularized political and social pattern, rather than the traditional Chinese concept of the continuity of heaven and man. Just as Li Ling 李零 said: "Literally, *Juedi Tiantong* should be called the division of *Tian* and man. It was



strange to translate '*jue*' 絶 (isolation) into '*he*' 合 (unity)" (L. Li 2000, p. 13). However, in fact, what *Juedi Tiantong* truly expressed was the exact process by which our Chinese ancestors clearly distinguished between nature and civilization, while still maintaining a vague relationship between the two. Just as *Zhouyi, Bi Gua* 周易·賁卦 (*The Zhou Book of Change, The Bi Hexagram*) said: "we observe the ornamental figures of the sky, and thereby ascertain the changes of the seasons. We look at the ornamental observances of the society, and understand how people are well educated" 觀乎天文, 以察時變; 觀乎人文, 以化成天下. On the one hand, our Chinese ancestors' construction of the social order had never been divorced from their understanding and imitation of the order of *Tiandao* 天道. On the other hand, *Juedi Tiantong* drew a clear boundary between man and gods, which meant that social and political affairs would be completely governed by humans themselves. This religious orientation encouraged Chinese philosophers to look for the root of value within human society rather than outside it, while *Tian* was still revered as a supreme god. This tradition is completely different from the values proclaimed by Western Christianity, where God is the only source of value.

The term nature was always absent in traditional Chinese philosophy, and the ancient Chinese people's understanding of nature was mainly reflected in the evolution of the relationship between *Tian* and man. The explanation of the relationship between *Tian* and man contained in *Juedi Tiantong* is conducive to a demonstration of the essence of the unity of heaven and man in ancient China and will provide a new perspective to understand the relationship between man and nature among different civilizations. Based on previous research outcomes, this article will conduct analysis from the perspective of conceptual history and philosophy, rather than regarding *Juedi Tiantong* as a purely historical event, which can better clarify the link between China's ancient shamanism and Confucianism. This article aims to analyze the impact of *Juedi Tiantong* on the relationship between *Tian* and humans in ancient China, in order to shed light on how *Juedi Tiantong* influenced ancient Chinese people's understanding of nature, and explore the reason why the term Ziran 自然 (nature) in Chinese philosophy is different from the term nature in modern Western ecological civilization. This article is mainly divided into five parts: the first part is the introduction, including a basic background on *Juedi Tiantong.* In the second part, the evolution of the concept of *Tian* will be interpreted. The third part will further explain the transformation of the mode of communication between *Tian* and man after *Juedi Tiantong*, in order to interpret how the communication mode changed from the original religious link to ethical and philosophical unity. The fourth part will discuss how *Juedi Tiantong* preserved the key elements of shamanic culture, which resulted in the absence of "nature" in traditional Chinese philosophy. Lastly, the conclusion will be explained.

## 2. *Juedi Tiantong* and the Metaphysical *Tian*

The reason why there are various explanations of *Tianrenheyi* 天人合一 (the unity of heaven and man) is the ambiguity concerning the connotation of *Tian*. In *Shuowen Jiezi* 説文解字 (*Annotations on the Chinese Characters*), *Tian* is interpreted as "the supreme". It can be seen that *Tian* originally referred to the largest and the highest of something. However, different philosophers interpret *Tian* according to different dimensions.[1] One of the most important effects of *Juedi Tiantong* on the relationship between *Tian* and man was the shaping of the metaphysical dimension of *Tian*. This section will analyze how *Juedi Tiantong* made *Tian* metaphysical.

### 2.1. *Juedi Tiantong* as a Key Event in Conceptual History

The term *Juedi Tiantong* was first mentioned in the chapter *Lüxing* 呂刑 (*The Marquis of Lü on Punishments*) in the *Shangshu* 尚書 (*The Book of Documents*), and other literature on this event included *Guoyu Chuyu* 國語·楚語 (*The Discourses of the States, The Discourses of the State of Chu*) and *The Classic of Mountains and Seas* 山海經. In fact, *Juedi Tiantong* was not a simple concept, but a religious reform lasting hundreds of years, which had exerted an important influence on Chinese conceptual history. It is said that *Lüxing* was written at the

time of King Mu of Zhou 周穆王 (r. 976–922BC), but the actual time of writing of the book is still unclear. According to Gu Jiegang's 顧頡剛 deduction, *Lüxing* was written at a time when the belief in gods was prevalent, while *Chuyu* was written at a time when the belief in gods was less popular (Gu and Liu 2005, pp. 1950–51). Therefore, logically, *Lüxing* was written first.

However, if we analyze *Juedi Tiantong* purely from the perspective of history, we are bound to enter into a dilemma. It is difficult to prove the historical authority of that ancient period. Huang Yushun 黃玉順 once suggested that this conceptual event marked the initial construction of Chinese metaphysics in the axial period (Huang 2005, pp. 8–11). Indeed, it is more reasonable to analyze *Juedi Tiantong* from the perspective of conceptual history, and the establishment of metaphysics[2] is closely related to the change in the connotation of *Tian*. *Lüxing* states that:

> According to the teachings of ancient times, Chi You 蚩尤 was the first to produce disorder, which spread among the quiet, orderly people, till all became robbers and murderers, owl-like and yet self-complacent in their conduct, traitors and villains, snatching and filching, dissemblers and oppressors. Among the people of Miao, they did not use the power of goodness, but the restraint of punishments. They made the five punishments engines of oppression, calling them the laws. They slaughtered the innocent, and were the first also to go to excess in cutting off the nose, cutting off the ears, castration, and branding. All who became liable to those punishments were dealt with without distinction, no difference being made in favor of those who could offer some excuse. The people were gradually affected by this state of things, and became dark and disorderly. Their hearts were no more set on good faith, but they violated their oaths and covenants. The multitudes who suffered from the oppressive terrors, and were (in danger of) being murdered, declared their innocence to Heaven. God surveyed the people, and there was no fragrance of virtue arising from them, but the rank odor of their (cruel) punishments. The great Emperor compassionated the innocent multitudes that were (in danger of) being murdered, and made the oppressors fell the terrors of his majesty. He restrained and (finally) extinguished the people of Miao, so that they should not continue to future generations. Then he commissioned Zhong and Li to make an end of the communications between earth and heaven; and the descents (of spirits) ceased. From the princes down to the inferior officers, all helped with clear intelligence (the spread of) the regular principles of duty, and the solitary and widows were no longer overlooked (translated by Legge 2013, p. 369).

According to this statement, the cause of *Juedi Tiantong* was the destruction of the social order, namely the political war caused by Chiyou's tribe. In order to survive, people prayed to *Shangdi* 上帝 for help. In this context, *Shangdi* announced a religion ban, which separated *Tian* 天 (the world of the gods) and *Di* 地 (human society) from each other. In other words, *Juedi Tiantong* caused a change, from the mixture of humans and gods (minshen zarou 民神雜糅) to the separation of humans and gods (*renshenxiangfen* 人神相分). *Juedi Tiantong* acclaimed the absolute boundary between *Tian* and humans, which contributed to the formation of the metaphysical connotation of *Tian*.

### 2.2. The Integration of Tian and Shangdi

From the perspective of conceptual history, *Juedi Tiantong* reflects the transformation from the primitive polytheistic belief system to a religion with a supreme god. In this process, *Tian* 天 gradually intermingled with *Shangdi* 上帝 and became a supreme god. Originally, *Tian* referred to the world of ghosts and gods in general, which also meant that there was no hierarchy of the gods in ancient people's thoughts. Chen Lai 陳來 also said that the most ancient pattern of sacrifice was probably the worship of power or objects in nature (L. Chen 2002, pp. 18–24). But after *Juedi Tiantong*, kingship in the human world with absolute political authority occurred. Correspondingly, *Tian* became the supreme god

(*Di* 帝), exercising control over all other gods. Chen Yun 陳贇 pointed out that, "In fact, the analysis of *Juedi Tiantong*, to some extent, involved the relationship between Shen 神 (gods) and *Di* 帝. On the one hand, Shen 神 were actually the ancestral gods of the various tribes and they were also the shamans with highest privilege in the tribe—the kings. On the other hand, *Di* 帝 had the higher position than ancestral gods, and *Di* was the only person eligible to sacrifice to *Tian*, namely the emperor in a later dynasty. The political and martial wars among tribes and states were ended by the construction of a hierarchy of gods" (Y. Chen 2010, pp. 16–23). Before *Juedi Tiantong*, people and gods were mixed, and all the people were able to communicate with the gods through shamans. At this time, the gods were diversified, and there was no absolute supreme god. After *Juedi Tiantong*, the communication between humans and gods had been cut off by the king, and political power coincided exactly with religious power.

The literal meaning of *Tian* is the physical sky above the earth, but from the inscriptions on bones from the Shang dynasty, it is obvious that *Tian* became a target of worship. Chao Fulin 晁福林 argued that *Tian*, during the Shang dynasty, was represented by *Di*, and *Tian* was a synonym of *Di* (Chao 2016, pp. 130–46). Jana S. Rošker said: "In the Shang Dynasty, *Tian* became the supreme deity of the state religion, and this did not change significantly until the period marking the transition from the Western to the Eastern Zhou Dynasty (eighth century B.C.)" (Rošker 2023). Furthermore, *Tian* during the Shang dynasty was likely to be a combination of natural gods and ancestral gods. "The highest god is transformed from the gods closely related to natural life" (Lü 1994, p. 664). Guo Moruo 郭沫若 once said: "the Yin people believed that *Tian* and *Di* were the supreme god with human emotions, who were able to make decisions according to its own preferences" (Guo 2005, p. 7). Such belief in *Tian* was adopted by Zhou from the Shang. *Tian* did not lose its position as a supreme god until the Spring and Autumn period. According to Yu Yingshi 余英時, the real target of China's axial breakthrough was the shaman culture rather than the culture of rituals and music. After the breakthrough, the various emerging schools of thought during the Spring and Autumn period and the Warring States period constructed a very different *Tian*, which was usually called Dao 道 (Y. Yu 2014, p. 32).

### 2.3. The Establishment of the Metaphysical Dimension of Tian

*Juedi Tiantong* also meant the establishment of the metaphysical dimension of *Tian*. Since then, *Tian* has become a form of absolute transcendence. According to Roger T. Ames, strict transcendence, in mainstream Western civilization, can be understood from the perspective of philosophy or theology. It asserts that an independent and superordinate principle A originates, determines and sustains B, where the reverse is not the case. Such transcendence renders B absolutely dependent upon A and, thus, it is nothing in itself (Ames 2016, p. 3). However, the relevance of such a strict form of transcendence to Chinese traditional philosophy is controversial. It is more appropriate to conceive *Tian* as a form of transcendence within the context of Confucian ethics, which has weakened the personality of *Tian* as a transcendent god but has partly preserved the sacredness of *Tian*. The reason why this metaphysical transition occurred was that *Juedi Tiantong* changed the mixture of humans and gods to a separation of the two and placed *Tian* far away from the human world. Political leaders monopolized the right to sacrifice themselves to *Tian* by staging a religious revolution, making *Tian* exclusive to political leaders and isolated from civilians. A consequence of the religious reform was the weakening of the personality of *Tian*. As Yao Zhongqiu 姚中秋 claimed *Tian* represented total supremacy over the others. All the gods were dominated by *Tian*. However, civilians were not allowed to talk to or hear from *Tian*. Zhuanxu made sacrifices to the silent and speechless *Tian*, shaping the fundamental features of the subsequent religious system in China. "The personality of the gods in China was weak, and even *Tian* was depersonalized" (Yao 2022, pp. 59–68). Before *Juedi Tiantong*, people believed that gods could talk, and shamans were the mediators, who were able to convey the instructions of the gods. The gods played a critical role in guiding people's daily lives. However, after *Juedi Tiantong*, the personality of *Tian* had declined

and the religious attributes of *Tian* were replaced by morality. Wang Ka 王卡 claims that *Tian* should be regarded as the nominal god (K. Wang 2016, pp. 23–26).

The victory of Zhou over Yin resulted in the rethinking of the worship of *Tian* among politicians in the early Zhou period. The Duke of Zhou (周公) proposed the theory of matching *Tian* with moral virtues (*Yidepeitian* 以德配天) to explain the transition of the Mandate of Heaven from Yin to Zhou. The Duke of Zhou said "Oh! God (dwelling in) the great heavens has changed his decree respecting his great son and the great dynasty of Yin. Our king has received that decree. Unbounded is the happiness connected with it, and unbounded is the anxiety: Oh! how can he be other than reverent?" (Shangshu, Zhaogao, translated by Legge 2013, p. 255) 嗚呼！皇天上帝，改厥元子茲大國殷之命，惟王受命，無疆惟休，亦無疆惟恤 (尚書·召誥).

The Duke of Zhou announced the royal will to the officers of the Shang dynasty, saying: "The king speaks to this effect: — 'Ye numerous officers who remain from the dynasty of Yin, great ruin came down on Yin from the cessation of forbearance in compassionate Heaven, and we, the lords of Zhou, received its favoring decree. We felt charged with its bright terrors, carried out the punishments which kings inflict, rightly disposed of the appointment of Yin, and finished (the work of) God" (Shangshu, Duoshi, translated by Legge 2013, p. 275) 爾殷遺多士！弗吊旻天，大降喪于殷。我有周佑命，將天明威，致王罰，敕殷命，終于帝 (尚書·多士).

It can be seen that the Duke of Zhou did not deny the absolute authority of *Tian*, and the victory of Zhou over Yin was closely related to the transformation of the Mandate of Heaven from Yin to Zhou. Moreover, the Duke of Zhou stressed that "The fact simply was, that, for want of the virtue of reverence, the decree in its favor permanently fell to the ground" (Shangshu, Zhaogao, translated by Legge 2013, p. 259) 惟不敬厥德,乃早墜厥命 (尚書·召誥). Apparently, the people in the Zhou dynasty believed that *Tian* would deprive the king without virtue of his authority. Religion began to be linked with morality.

Then, in the Spring and Autumn period and the Warring States period, with a strong sense of responsibility for preserving the culture of the Zhou dynasty, Confucius kept parts of the transcendent religious feature of *Tian*, but, at the same time, did not regard *Tian* as a kind of external power influencing the practices of nature. On the one hand, Confucius showed great reverence and awe in regard to *Tian* and the Mandate of Heaven. Confucius said, "He who does not understand the will of Heaven cannot be regarded as a gentleman" (The Analects, Yao Yue, translated by Waley 2008, p. 233) 不知命，無以為君子也 (論語·堯曰). On the other hand, Confucius regarded *Tian* as a speechless supreme god, without a strong personality. Confucius said, "Heaven does not speak; yet the four seasons run their course thereby, the hundred creatures, each after its kind, are born thereby. Heaven does no speaking!" (The Analects, Yang Huo, Translated by Waley 2008, p. 205) 天何言哉？四時行焉，百物生焉，天何言哉？《論語·陽貨》. *Tian* dominated the alternation of the four seasons and the birth and termination of all creatures, but *Tian* was always silent. Therefore, it is more reasonable to regard *Tian* in Confucius' thoughts as a metaphysical conception, rather than a transcendent god. Just as Tang Junyi 唐君毅 said, "The Chinese as a people have not embraced a concept of "Heaven" (tian 天) that has transcendent meaning. The pervasive idea that Chinese people have with respect to tian is that it is inseparable from the world" (Tang 1991, p. 241).

In a word, after *Juedi Tiantong*, the connotation of *Tian* changed greatly. In ancient times, there was no supreme god, and the unity of heaven and man meant the consistency between the gods and the human world with the help of shamans. But after *Juedi Tiantong*, *Tian* became a silent supreme god and gradually referred to metaphysical principles, and the unity of heaven and man refers to the continuity between human nature and *Tiandao* 天道.

## 3. The Change in the Communication Mode between *Tian* and Man

If the character *jue* 絕 can be understood as a verb meaning to monopolize or to cut off, then before the completion of *jue*, there existed a period allowing the communication

between *Tian* and man; and, after *jue*, a new pattern was formed, in which man could not communicate with *Tian* at will. As a revolution in conceptual history, in addition to the change in the meaning of *Tian* mentioned above, *Juedi Tiantong* also indicated two different communication modes between *Tian* and man. Before *jue*, man frequently established a link with the gods through religious ceremonies and man was submitted to the gods. While after *jue*, the connection between *Tian* and man was mainly emphasized by Chinese ethical philosophy, where humans focused on the self-awareness of their own subjectivity. The former was the religious and mythical form of communication between *Tian* and man, which can be called *Tong* 通. The latter was the humanistic and rational form of communication, which should be called *He* 合. In this part, the two different communication modes between *Tian* and man will be analyzed, in order to shed light on how *Juedi Tiantong* facilitated the transformation from Chinese shamanism to Confucianism.

### 3.1. Tong 通: The Religious and Mythical Form of Communication between Tian and Man

China's ancient religion was a typical form of shamanism, in which shamans were the mediators between gods and man. This period basically corresponds to the stage before *Juedi Tiantong*, when gods and man were mixed (*minshen zarou* 民神雜糅) and everyone could become a shaman and historian (*Jiawei Wushi* 家為巫史). At this time, human agricultural activities were still greatly influenced by natural conditions, and human beings were subjugated to the will of natural gods. People believed that they could receive instructions from the gods through religious practices, and the connotation of *Tong* 通 provided the exact evidence of this.

In modern semantics, *Tong* 通 means to get from one place to another without obstacles. *Shuowen Jiezi* 說文解字 also said that *Tong* means "to reach". However, in ancient times, the context of using *Tong* 通 was closely linked to religious rituals. Originally, *Tong* was one of the important alternatives to *Sheng* 聖 (sacredness), which was initially synonymous with *Ting* 聽 (hear) and was also cognate with *Sheng* 聲 (voice). Guo Moruo 郭沫若 explained that in ancient times, the character of *Ting, Sheng, Sheng* 聽聲聖 were actually the same. *Ting* 聽 consisted of a mouth and ear. The mouth means that there is something to say and the ear means listening. What people intended to listen to was the instructions from the gods and people skilled in listening to the gods' instructions were called *Sheng* 聖 (Guo 2002, p. 489). Zhu Junsheng 朱駿聲 said that the so-called *Shengren* 聖人 (sages), before the Spring and Autumn period, were also known as *Tongren* 通人 (shamans). After the Warring States period, *Shengren* 聖人 (saints) specifically referred to the people with perfect moral virtues (Zhu 1983, p. 872). Li Xiaoding 李孝定 also claimed that *Sheng* 聖 originally referred to a person who had good auditory perceptions and *Tong* 通 was a derivate of *Sheng* 聖 (X. Li 1965, p. 3519). Therefore, we can deduce that *Tong* 通 was related to mental activities and implied a complex spiritual experience that occurred during religious rituals. In this case, in addition to meaning physically unobstructed roads, *Tong* 通 also meant the path by which humans could have access to the gods. Zhao Jiang 趙江 concluded that *Tong* 通 can be understood in regard to three dimensions: (1) the communication and connection between *Tian* and man; (2) the communication between political leaders and civilians; and (3) to be purely skilled or proficient in a certain skill (J. Zhao 2023, p. 99–102).

The original meaning of *Tong* 通 shows that, according to the perception of the ancients, the communication between man and the gods could be real. Shamans were able to meet or talk to the gods through rituals or sacrifices and even invite the gods to descend to the human world. In this period, the communication between man and *Tian* was usually analyzed within the field of religion. *The Classic of Mountains and Seas* records some specific communication methods. Previously, some scholars believed that, with the inclusion of ridiculous and absurd words, *The Classic of Mountains and Seas* was a collection of mythological stories. Furthermore, it was difficult to certify the historical authenticity of the events recorded in the book. However, it is undeniable that the creation of myths can hardly be groundless. In fact, how ancient people recognized the world is just hidden within these so-called illogical words. The connection between *Tian* and man may not be a

real historical experience, but it can be a real conceptual form of existence. The ancient people's understanding of *Tian*, namely the spirits, ghosts and gods presented in *The Classic of Mountains and Seas* was the most primitive explanation of the situation of human beings in the universe, at a time when ancient people could not think rationally and get rid of the effect of religion. *The Classic of the Great Wilderness*: *The Western* (Volume 16 of *The Classic of Mountains and Seas*) 山海經·大荒西經 states that:

> Inside the Great Wilderness there is a mountain called *Mount of the Sun and the Moon* 日月山 which is the pivot of the sky. *Wujuntianmen* 吳姖天門 is where the sun and the moon set. There is a god who has a human face and no arms. With two feet bent reversely onto his head, he is called Xu 噓. King Zhuanxu gave birth to Laotong 老童. Laotong gave birth to Chong and Li. The God of Heaven ordered Chong to hold up the sky and Li to press down the earth. After finishing his job, Li gave birth to Ye 噎. Then Li lives at the West Pole, presiding over the movement of the sun, the moon and the stars (translated by Wang and Zhao 2010, p. 295).

The text describes a creation myth. Just as Liu Zongdi 劉宗迪 said, heaven was mingled with the earth originally. It was the two brothers, Chong and Li, who worked together in the effort to lift up and press down, which made the heaven and the earth separate from each other (Z. Liu 2020, pp. 64–71). Therefore, Chong, Li and Ye were the gods of creation, who established the order of space and time. Yang Kuan 楊寬 also believed that *Juedi Tiantong* was the myth of creation, which was passed down from the Western Zhou dynasty to the Warring States period (K. Yang 1999, p. 832). From the perspective of modern physical science, the reason for the formation of the universe was certainly not due to the intervention of human or divine power, but *Juedi Tiantong* reflects the ancients' understanding of the relationship between heaven, earth, gods and human beings. Ancient people believed that heaven and earth were close to each other, and it was not difficult for people to ascend to heaven or for gods to descend to earth. In ancient times, when humans were incapable of thinking logically and rationally, they tended to believe that there was a great correlation between human activities and the gods of nature. At this time, the relationship between *Tian* and man was a primitive unity of man and nature, and human beings were not able to establish self-awareness.

The "pivot of the sky" in this text shows us a more concrete way of communicating between heaven and man. The pivot of the sky represents the gate of heaven, through which humans and gods can meet with each other. Just as Gong Zizhen 龔自珍 claimed that at the very beginning, man could ascend to heaven and the interaction between man and heaven occurred all the time (Gong 1975, p. 13). This is also evidenced in the question by King Zhao of Chu 楚昭王 in *Guoyu Chuyu* that if *Chong* 重 and *Li* 黎 did not make heaven and earth inaccessible, would the people be able to ascend to heaven? The phrase "ascend to heaven" suggests that there was a concrete path between heaven and earth, through which people could indeed climb up to heaven to meet the gods. The meeting location was often famous peaks, such as the *Mount of the Sun and the Moon* 日月山. Through the long-term observation of natural phenomena, ancient people found that the sun, the moon and stars in the sky seemed to rise from certain mountains and also fell down onto them. Therefore, they believed that the gods in heaven could descend to earth and live in these mountains.[3] This was, of course, the most primitive form of imagination, but such imagination made the sun, the moon, the stars and mountains sacred. Most mountains in *The Classic of Mountains and Seas* were places of worship.

*The Classic of Mountains and Seas* also records a large number of mountains, which provided access to heaven, such as *Dengbao Mountain* 登葆山. This shows that ancient people believed that towering mountains were one of the paths for people and gods to communicate with each other. However, after *Juedi Tiantong*, except *Zhuanxu, Chong* and some other designated shamans, other people were no longer allowed to have contact with the gods. Xu Xusheng 徐旭生 further pointed out that the specific implementation process of the severance was to close the path to climb, so people could not climb as they pleased

(Xu 2023, pp. 124–25). Liu Zongdi 劉宗迪 also deduced that a complete description of the story of *Juedi Tiantong* should be as follows: the two brothers, *Chong* and *Li*, collaborated with each other to separate the heaven and the earth and left the only access to heaven on *the Mountain of the Sun and the Moon*. Then, the two brothers became the guards of that access (Z. Liu 2020, pp. 64–71).

It can be seen that before *Juedi Tiantong*, our ancestors' perception of the natural world was always entangled with the imagination of the gods. It was only after *Juedi Tiantong* that the boundaries between humans and heaven became clear and distinct. Then, it was possible for human beings to get rid of the interference by the gods and begin to discover the instinctive value of human beings. As Yin Rongfang 尹榮方 said, many nations had myths of creation with a common cultural implication being that the end of the age of chaos was marked by humanity's understanding and realization of time and space and the creation of calendars (Yin 2012, pp. 232–39). When human beings discovered fixed and unchanging natural laws, they gradually got rid of primitive obscurantism. This was what *Juedi Tiantong* depicted in *The Classic of Mountains and Seas*. People could no longer measure time or communicate with the gods privately, and only the rulers could enact the calendar and establish a unified social order. *Juedi Tiantong* can be regarded as the termination of human–god relations, after which, gods returned to gods and people returned to people.

### 3.2. He 合: The Humanistic and Rational Form of Communication between Tian and Man

If *The Classic of Mountains and Seas* suggests the most simple and primitive way of communication between heaven and man, then the communication mode described in *Shangshu*, *Lüxing* and *Guoyu Chuyu* has been changed to *He* 合 between *Tian* and man, which appeared after *Juedi Tiantong*. This can be clearly interpreted from Guanshefu's 觀射父 explanation to King Zhao of Chu, which was recorded in *Guoyu Chuyu*:

> Anciently, men and spirits did not intermingle. At that time there were certain persons who were so perspicacious, single-minded, and reverential that their understanding enabled them to make meaningful collation of what lies above and below, and their insight to illumine what is distant and profound. Therefore the spirits would descend into them. The possessors of such powers were, if men, called xi 覡 (shamans), and, if women, wu 巫 (shanmanesses). It is they who supervised the positions of the spirits at the ceremonies, sacrificed to them, and otherwise handled religious matters. As a consequence, the spheres of the divine and the profane were kept distinct. The spirits send down blessings on the people, and accepted from them their offerings. There were no natural calamities (paraphrased by Bodde 1981, pp. 45–84).

First of all, we need to know that the sentence "anciently, men and spirits did not intermingle" does not concern the most primitive stage of religion. As mentioned above, ancient people believed that there was no clear boundary between man and heaven, and anyone could become a shaman. The institutional construction of the four types of ritual specialists and the five kinds of officials in Guanshefu's 觀射父 statement was based on the highly mature religious and cultural system of the Western Zhou dynasty (D. Yu 2005, p. 12). Ritual specialists did not emerge until the Western Zhou dynasty, when the personality of *Tian* had already declined, and the humanistic rationality hidden within *Tian* gradually became dominant.

Primitive shamans relied on their individual talent to invite the gods to descend to meet them, but the main obligation of ritual specialists, as described by Guanshefu 觀射父, was to make administrative arrangements for sacrifices. Ritual specialists were only professionals who were well versed in the protocols of the sacrificial rituals. The emphasis of Guanshefu's 觀射父 description of ritual specialists was not to manifest the sanctity and mystery of religion, but rather to make shamans disenchanted by emphasizing the extremely high level of morality that ritual specialists had to possess in order to be qualified. The ultimate function of ritual specialists was not to convey the instructions from *Tian* to the people on earth, but to cultivate people's virtue and loyalty and lead all people

to behave morally. In this case, *Tian* was regarded as a symbol of justice and impartiality (Lu 2010, pp. 162–70). At this time, the function of *Tian* as a silent personal god had been narrowed down to monitoring and guiding the behavior of the emperor, rather than giving direct instructions to the people. All this shows that during the Spring and Autumn period and the Warring States period, *Tian* moved from the religious field into the field of humanity and rationality. Confucianism, pre-Qin dynasty, did not completely deny the existence of *Tian*, but the focus of *Tian* was on secular affairs and humanity's efforts rather than on divinity. After *Juedi Tiantong*, China's primitive shamanism gradually transformed into a moral and ethical religion. Guanshefu's philosophical and rational answer to the mythological question by King Zhao of Chu was undoubtedly linked to a kind of humanistic guidance to both the king and the people, which ultimately resulted in the attributes of the Chinese *Tian* being completely different from that of Western religions. Yao Zhongqiu 姚中秋 proposes that the relationship between *Tian* 天 (heaven) and *Ren* 人 (man) in China is different from the relationship between personalized gods and human beings in the West. The Chinese people achieved a revolutionary breakthrough in regard to humanization and rationalization, which may be the earliest occurrence of such a breakthrough in all human civilization. Yao further emphasizes the silence and speechlessness of *Tian* after *Juedi Tiantong*. Since *Tian* does not speak, the communication between *Tian* and man no longer relies on a face-to-face meeting (Yao 2022, pp. 59–68). After *Juedi Tiantong*, the sacred character *Tong* 通 evolved into the secular and philosophical character *He* 合 (the conformity between *Xinxing* 心性 and *Dao* 道).

To summarize, in ancient times, the method of communication between *Tian* and man included the most simple form of physical communication, such as meeting on towering mountains. Such a religious mode of communication can be called *Tong* 通. After *Juedi Tiantong*, only the king, due to their political and social privilege, could be the mediator between *Tian* and man, and religious sacrifices to *Tian* could only be carried out by the king and ritual specialists. *Tian* turned into a silent and impersonal supreme god, with moral metaphysical characteristics. The communication between *Tian* 天 and *Ren* 人 no longer occurred through religious sacrifices, but moved into the realm of ethics and morality, namely the compatibility between *Xinxing* 心性 (human nature) and *Tiandao* 天道 (the principles of nature). What philosophers during the Spring and Autumn period and the Warring States period were concerned with was the ultimate justification for human civilization, and whether the casting of human nature was in conformity with *Tiandao* 天道. At this time, *Tian* 天 metaphorically referred to the rationality generally existing in everything, and it concerned the ideological existence about the commonality and rationality of all things (Lu 1998, p. 1).

## 4. *Juedi Tiantong* and the Abstraction of Nature

As an important religious revolution, *Juedi Tiantong* not only transformed the meaning of *Tian* and the way in which *Tian* and man communicated with each other, but also influenced man's understanding of nature, which was mainly reflected in China's ancient astrology and the reform of the calendar. In a contemporary context, the word nature generally refers to objects in nature, such as the sky, stars, earth, mountains and rivers. However, in Chinese philosophy, *Ziran* 自然 (nature) does not mean the natural world, and *Tian*, as one of the most important concepts in Chinese philosophy, is usually a metaphysical concept. *Juedi Tiantong* abstracted the term *Tian*, which originally meant concrete and physical natural objects, into a metaphysical concept.

### 4.1. *Juedi Tiantong* and the Revolution of the Calendar

In ancient China, the enactment of the calendar depended on the observation of natural phenomena; therefore, changes in the calendar largely reflected the ancient people's understanding of the natural world. In ancient times in China, astronomy, astrology, the calendar and religious rituals were closely related to politics, so *Juedi Tiantong* was not only a revolution in religion, but also an important revolution in regard to astronomy and the

calendar (Wu 2023, pp. 1–8). According to Yin Rongfang 尹榮方, the *Mount of the Sun and the Moon* was actually the astronomical observatory in ancient times, and Chong and Li were the two divine beings living in the pivot of the sky, responsible for connecting *Tian* and man (Yin 2012, p. 229). Their duty was to observe the sun, moon and stars and to formulate the calendar according to the seasonal changes, in order to guide agricultural production, which was called *Guanxiang Shoushi* 觀象授時. In fact, during the Han dynasty, explaining *Juedi Tiantong* from an astronomical perspective was beyond doubt. For example, *Shiji*, *Lishu* 史記·曆書 (*Records of the Historian, The Book of Calendar*) recorded that Huangdi 黃帝, Zhuanxu and the families of Chong and Li were all specialists in astronomy and made outstanding contributions to formulating the calendar. What *Juedi Tiantong* expressed was that Chong and Li mastered a more accurate method of stargazing and monopolized the interpretation of God's will by formulating a premier calendar that made agriculture more efficient. Consequently, the general public could no longer privately determine the calendar or sacrifice themselves to *Tian*.

It is especially important to note that in ancient China, the observation of the stars and the measurement of time were not a purely scientific form of study. The ancient people's exploration of nature was always linked to the belief in *Tian* and the worship of the gods. In ancient China, science and religion were intermingled. The observation and recording of stars were scientific and rational, but the explanation of the principles of movement was religion-based. It was difficult for Chinese ancients to interpret the changes in nature without the help of the gods. According to Liu Zongdi 劉宗迪, the starry sky was regarded as the origin of divinity, the dwelling pale of the gods and the source of the Mandate of Heaven. Therefore, for the ancients, astronomy was not only a kind of scientific knowledge, but also the manifestation of *Tian* and divinity (Z. Liu 2020, pp. 64–71). It is only in the field of religion and astronomy that we can understand that the ancient people's awe and worship of the natural world may have come from the sense of grandeur and vastness that they experienced when they looked up at the stars at night.

The best archaeological proof of this is the site of the *Tao Temple* 陶寺 in Xiangfen, Shanxi 山西襄汾, a large complex used for measuring time and performing rituals.[4] It proved that the measurement of time was the most direct way for ancient people to communicate with the gods, and observing the starry sky was the only way to understand the deity (Z. Liu 2016, pp. 1–9). It is not difficult to see that humanity and astronomy, science and religion, and rationality and divinity were inseparable in ancient people's view of nature. In this regard, it was difficult for ancient Chinese philosophers to formulate natural philosophy centered on "logos" like ancient Greek philosophers. It is also difficult for Chinese philosophers to recognize *Tian* as nature.

### 4.2. The Abstraction of Nature

Chinese philosophy advocates *Tianren Heyi* 天人合一 (the unity of heaven and man). However, *Tian* 天 (heaven) is not the equivalent of *Ziran* 自然 (nature). In other words, the concept of *Ziran* 自然 (nature), broadly used in Chinese philosophy, is fundamentally different from the term "nature" in Western ecological philosophy. As Zhang Rulun 張汝倫 said, due to the influence of modern Western civilization, people gradually re-understand the relationship between *Tian* and man according to modern Western realism. *Tian* is understood as "physical nature", and man is understood as "subjectivity". The consequence of such a misunderstanding is the deviation from the tradition of Chinese philosophy (R. Zhang 2019, pp. 52–58). Zhang's assertion is basically correct; however, the problem is why *Tian* 天 (heaven) in Chinese philosophy cannot be understood as physical nature?

In response to this question, *Juedi Tiantong* reminds us of an important clue: that the ancient Chinese people's observation and cognition of natural phenomena have always been associated with religion. Before *Juedi Tiantong*, our ancient ancestors believed that the gods and man were intermingled. Human beings could not completely distinguish themselves from nature. The totemic belief found in many ancient nations is an example, and the boundary between man and heaven was not clear. After *Juedi Tiantong, Tian* moved

from the field of religion into the realm of humanism and ethics. However, "nature", originally contained in *Tian*, did not become an independent object to be understood and studied. Furthermore, after the breakthrough of philosophy during the Spring and Autumn period and the Warring States period, *Tian* became an abstract concept referring to the principle that universally exists in all things. *Tao Te Ching* 道德經 (part 1) said that "Man takes his law from the Earth; the Earth takes its law from Heaven; Heaven takes its law from the Dao. The law of the Dao is its being what it is" (translated by Legge 2021, p. 17) 人法地，地法天，天法道，道法自然. Originally, *Tian* was regarded as the supremacy in the world, but Daoism added Dao on the top of *Tian*. In addition, Daoism established an abstract general principle, Tao, which relates to allowing everything to be themselves. In this case, *Ziran* 自然 (nature) is the abstract law of *Tian*, which keeps the universe moving, and no longer refers to the physical natural world. Confucianism also focuses on the moral attributes of *Tian*, emphasizing the capability of man to be in harmony with *Tian* in terms of moral virtue, rather than the harmonious coexistence between man and ecological nature. Just as *Zhouyi Qiangua* 周易·乾卦 said, a great man has virtue vast as heaven and earth and wisdom as brilliant as the sun and the moon. He works in pursuit of the good order as the alteration of the seasons and reveals good fortune and disaster in his miraculous divination as ghosts and spirits do 夫大人者，與天地合其德，与日月合其明，與四時合其序.

## 5. Conclusions

To summarize, *Juedi Tiantong*, an important religious revolution in ancient China, not only shaped the metaphysical conception of *Tian* in Chinese philosophy, but also laid down the main mode of communication between *Tian* and man, which was the religious foundation of the unity of heaven and man. Karl Jaspers said that despite being likewise bound, man brings forth his environment in a boundless overpassing of his ties. Life in an environment that he has created himself, simultaneously with life in the natural environment, is the hallmark of humanity (Jaspers 1965, p. 101). The formation of civilization depends on a certain natural environment, but the environment is not the only decisive factor. In fact, we find that religion also affects the relationship between human beings and the natural environment in reverse. Just as Pan Zhichang 潘知常 said, as the dominate and leading value in the axial era, religion undoubtedly had a historical rationality, because religion represented the first supernatural and fundamental value constructed by human beings, and also represented the initial self-awareness of humanity. Life was no longer natural and finite, but spiritual and infinite (Pan 2023, pp. 73–83). The significance of religion lies in the transcendence of nature. In this respect, the differences between Chinese and Western civilizations in the axial age were not only due to the various geographic conditions, but also due to the diverse religious cultures. In fact, the inward path of transcendence[5] in Chinese philosophy formed in the axial age actually originated from religious culture, especially the transcendent tradition formed by *Juedi Tiantong*. The severance between *Tian* and man after *Juedi Tiantong* shaped the metaphysical dimension of *Tian* and laid the foundation for the later Confucianism to pursue the intrinsic continuity and inseparability of human nature and the cosmic order (天道 *Tiandao*), and formed the unity of heaven and man with the core value of spiritual cultivation. Although the unity of heaven and man can hardly be understood as a simple harmonious coexistence between man and nature, this value orientation implies people's awe and respect for nature, and should be highly emphasized in modern times.

**Author Contributions:** Conceptualization, Z.T. and X.L.; methodology, Z.T.; software, X.L.; validation, Z.T. and X.L.; formal analysis, X.L.; investigation, Z.T.; resources, Z.T.; writing—original draft preparation, Z.T. and X.L.; writing—review and editing, X.L.; visualization, Z.T.; supervision, Z.T. All authors have read and agreed to the published version of the manuscript.

**Funding:** This research received no external funding.

**Institutional Review Board Statement:** Not applicable.

**Informed Consent Statement:** Not applicable.

**Data Availability Statement:** No new data were created or analyzed in this study. Data sharing is not applicable to this article.

**Conflicts of Interest:** The authors declare no conflict of interest.

## Notes

[1]   For example, Zhang Dainian 張岱年 believed that *Tian* has three meanings: the supreme master, nature and the most basic principle. Mou Zhongjian 牟鐘鑒 also believed that *Tian* has three meanings: the primordial existence, the natural existence and the transcendent existence. Ren Jiyu 任繼愈 proposed that there are five meanings of *Tian*: *Tian* of master, *Tian* of destiny, *Tian* of rationality, *Tian* of personality and *Tian* of nature.

[2]   According to Heidegger, philosophy is metaphysics. Metaphysics relates to being as a whole: the world, man, God with respect to being, with respect to the belonging together of beings in being. Metaphysics views beings as being in the manner of representational thinking which provides reasons. For since the beginning of philosophy and with that beginning, the being of beings has shown itself to be the ground (arche, aition). The ground is from where beings as such are what they are in regard to their becoming, perishing and persisting, as something that can be known, handled and worked upon. As the ground, being brings beings to their actual presence. The ground shows itself as a presence. (Heidegger 1972, pp. 55–56) Similarly, this final ground expressed in Chinese philosophy is the concept of *Tiandi* 天帝—*Tiandao* 天道—*Xinxing* 心性, which was established during the transformation of the axial period.

[3]   In *The Classic of the Mountains and the Seas*, a number of mountains in the human world are regarded as the places where gods dwell 帝之下都. For example, according to *The Classic of Western Mountains*: 400 li to the southwest is a mountain called Kunlun. This is god's dwelling place in the human world. *Luwu*, a god who has a tiger's body, nine tails, a human face and a tiger's paws, presides over this mountain. He is also in charge of the Nine Parts of the Sky and the God of Tian's Zoo for the Seasons.

[4]   There are a large number of articles about the important influence of the Tao Temple 陶寺 on China's ancient astronomy and calendar, such as Wang, Zhenzhong 王震中 2015 (Z. Wang 2015). *Tao Temple and Yao City—the Paradigm of China's Early States* 陶寺與堯都—中國早期國家的典型. *Southern Cultural Relic* 南方文物 3: 83–98.

[5]   Confucianism emphasizes the unity of human nature 人性 and *Tiandao* 天道. Confucius put forward the thought of "benevolence" 仁, which, to some extent, focuses on the subjectivity of morality and spirit. People have the ability to cultivate their inner morality, to achieve inner moral perfection, which is called "benevolence". In a word, the moral root of society is human subjectivity. Schwartz called such an inner turn to ethics the "inward transcendence" that brings a subjective focus to moral and spiritual life (Benjamin I. Schwartz 1975. Transcendence in ancient China. *Daedalus* 104: 57–68).

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
