# Peer review of "Juedi Tiantong: The Religious Basis of the Relationship between Tian and Man in Ancient China"

_religions, doi:10.3390/rel15040477_

Round 1
Reviewer 1 Report
Comments and Suggestions for Authors
This paper is not ready to be presented to peer reviewers. There are too many errors on the first few pages. The initial content makes no sense. I suspect this is a paper that was simply translated (poorly) into English. It was not written for an international audience. The author doesn't define the terminology under question: what exactly does Juedi Tiantong 绝地天通 mean? This is clear maybe to Sinologists and Chinese readers, but nobody else will figure this out.
The bibliography also doesn't consult much Western scholarship. If writing in English, why not cite scholarship written in English or other European languages? How does this study actually relate to wider questions in Sinology?
My notes:
This paper should use 繁體字, not 簡體字.
绝地天通 = 絕地天通
You need to translate 絕地天通 and make it comprehensible to readers unfamiliar with the concept.
"ancient national religion,"
What exactly is this? Was there even a nation before the 秦漢 period? This wording is confusing.
"Religion" is also a concept that is problematic in a premodern Chinese concept. They didn't use the word 宗教, which is a translation from European languages ("religion"). People before Buddhism's arrival in China didn't necessarily conceive of what they did as "religious" or "religion" in nature. So what exactly would a national religion mean in that period?
shaman 萨满 = why add 拼音 here?
"Zhou people 周人's conception" This is not formatted correctly. You don't add the plural marker ('s) to 漢字 like this. This paper is not properly edited.
"book of Changes 《周易》:"
Again, Book of Changes, not book of Changes. Also the title is 周易, so where the Zhou in the translation?
"Judging from astronomy to detect when the change, judging from the humanities into the world to"
This makes no sense at all.
I'm not reading further. This paper is not coherent or edited properly. I can't make sense of what is being said so far. The author needs to rewrite the whole paper.
Comments on the Quality of English LanguageThe paper was shown to editors, but it reads like a close translation. There are also numerous typos.
Author Response
Dear editor
Thank you for giving me the opportunity to submit a revised version of my paper. I really appreciate your time and effort to provide valuable suggestions for the revision of this paper.
Here is a Point-by-point response to review's comments and concerns
|
Comments 1: This paper is not ready to be presented to peer reviewers. There are too many errors on the first few pages. The initial content makes no sense. I suspect this is a paper that was simply translated (poorly) into English. |
|
Response 1: Thank you for pointing this out. We agree with this comment. Therefore, we have thoroughly checked the whole English translation and made substantial revisions including grammatic errors(line43,line93-95), vocabulary selection(line34-35, line113), sentence restructure(line 38-40, line84-86) and so on(the revised places will not be listed here one by one). However, we cannot agree with the reviewer’s remark that” The initial content makes no sense”. If the “initial content” refers to the introduction, then we argue that the introduction aims to clarify what kind of academic research has been made, what relevant questions about this topic have been proposed by scholars and how this article is organized. We have revised some grammatic problems in introduction and we hope the “initial content” is apprehensible at this time. The introduction begins with a brief explanation of Juedi Tiantong to give readers basic information on the meaning of this terminology, then the previous academic articles on Juedi Tiantong have been analyzed. So we added a contribution at the end of introduction to show what kind of questions this article aims to solve and how this article contributes in this field.(line114-125) The explanation of the relationship between Tian and man contained in Juedi Tiantong is conducive to the demonstration of the essence of the unity of heaven and man in ancient China, and will provide a new perspective to understand the relationship between man and nature among different civilizations. Based on the previous research outcomings, this article will conduct the analysis from the perspective of conceptual history and philosophy rather than regarding Juedi Tiantong as a pure historical event, which can better clarify the link between China’s ancient shamanism and Confucianism. |
|
Comments 2: It was not written for an international audience. The author doesn't define the terminology under question: what exactly does Juedi Tiantong 绝地天通 mean? This is clear maybe to Sinologists and Chinese readers, but nobody else will figure this out. |
|
Response 2: Firstly, we totally understand that, for the international readers not familiar with Chinese philosophy, it can be hard to understand the meaning of Juedi Tiantong. Therefore, in original manuscript, we gave a basic description to Juedi Tiantong in part 2.1”Juedi Tiantong as a thought event”(the subtitle of part 2.1 now has been revised as Juedi Tiantong as a Key Event in Conceptual History). Secondly, we completely agree that the definition of terminology is critical to understand the whole meaning of this article. Accordingly, we have added a brief explanation of Juedi Tiantong and also a historical background of this event, which can be found from line 21 to 32. However, we also argue that, this article mainly focuses on the discussion of the changes to the relationship between Tian and man brought by Juedi Tiantong and the reasons why this was unique to Chinese philosophy. This article is not hermeneutic or interpretive, and we assume that potential readers have basic understanding of shamanic culture in ancient China. A comprehensive explanation of Juedi Tiantong which may take a huge amount of words is not the focus of this article. In addition, Juedi Tiantong is regarded as a historical event lasting hundreds of years rather than a conception, and it is almost impossible to give a clear definition of it. Here is the newly added explanation which can be found at the very beginning of introduction (line21-32) Juedi Tiantong 絕地天通(the severance between Tian and man) was a religion re-form of great significance happened in ancient China. The exact time of this religion reform is still under debate among historians, but most scholars agree that Zhuanxu 顓頊, the leader of Juedi Tiantong, approximately lived during the period from 3000BC to 2500BC (Wang 2013, p.381). The main reason of this revolution was likely the war during the integration of tribal states at that time(Yu 2005,p.11-18). This was a religion revolution lasting hundreds of years, which changed the relationship between Tian天and man. Juedi Tiantong literally means the severance of the communication between Tian and man. Before Juedi Tiantong, all shamans were able to invite and talk to gods at any time. However, Zhuanxu, the martial and political leader with highest prestige at that time, announced a religion policy that only himself and a few designated shamans were allowed to communicate with Tian and all other people were prohibited from the sacrifice to Tian. Comments 3: The bibliography also doesn't consult much Western scholarship. If writing in English, why not cite scholarship written in English or other European languages? How does this study actually relate to wider questions in Sinology? Response 3: We totally understand the importance of citing more references written in European languages. Accordingly, we have looked up more relevant English articles and added two English-written reference.
1. Puett also focused on the transition of shamans into ritual specialists caused by Juedi Tiantong, he argued that spirits and humans should be separated and placed within a proper hierarchy of functions, and after Juedi Tiantong, Wu are not shamans at all (Puett 2022,p.107).(line73-76) 28. Puett, Michael J. 2022.To Become a God: Cosmology, Sacrifice, and Self-Divinization in Early China. Harvard University Asia Center.(line644) 2. Rošker said: “In the Shang Dynasty, Tian became the supreme deity of the state religion, and this did not change significantly until the period marking the transition from the Western to the Eastern Zhou Dynasty (eighth century BC)” (Rošker, 2023)(lin210-213) Rošker, Jana S. 2023. Models of Humanism in Ancient China: An Explanation Centered on Early Confucian Ethics. Religions 14, no. 1: 83. https://doi.org/10.3390/rel14010083 (line645-646)
However, what need to clarify is that Juedi Tiantong is completely a event happened in ancient China and all the original literature about it was written in classical Chinese. Most modern articles discussing Juedi Tiantong are written in Chinese and rarely in English. (However, this does not mean the study on Juedi Tiantong in English or other European language is not important. Quite the opposite, it is of great significance of publishing relevant articles in English in order to communicate with more international scholars and make cross-cultural study). We hope the added English references can make this article more easier to understand.
Comments 4:This paper should use 繁體字, not 簡體字.绝地天通 = 絕地天通 Response 4: Thank you for pointing this out. We agree with this comment. Accordingly, all the simple Chinese characters have been changed to traditional Chinese characters, and revisions can be found at: line 21, line 34, the whole bibliography and so on.(There are lots of traditional characters, so revisions will not be listed here one by one here).
Comments 5: ancient national religion," What exactly is this? Was there even a nation before the 秦漢 period? This wording is confusing. "Religion" is also a concept that is problematic in a premodern Chinese concept. They didn't use the word 宗教, which is a translation from European languages ("religion"). People before Buddhism's arrival in China didn't necessarily conceive of what they did as "religious" or "religion" in nature. So what exactly would a national religion mean in that period? Response 5: This is a very crucial comment to this article and thanks for pointing this out. Firstly, the meaning of nation varies with civilizations. Indeed, there was no united nations in the current Chinese geographical area before Qin dynasty. In this article, Juedi Tiantong approximately occurred during B.C 3000 to B.C 2500, and the influence of it lasted more than thousands of years. How to define the social organization pattern of this period in China is still a controversial topic in archaeology field. Some scholars use “early Chinese states”. Accordingly, we have changed “national” to the word “state”(in line 45) in order to make it clear to audience. Secondly, the translation of “religion” in Chinese and how to describe Chinese people’s beliefs before the arrival of Buddhism are still worth discussing. It may be not appropriate to use the word “religion” when we talking about ancient Chinese people’s belief, but undeniably, there were shamans, sacrifice and beliefs in gods and spirits before Buddhism's arrival in China and we still to need to find a word to show what we are talking about. Some scholars use the word shamanism. Therefore, in this article, before Juedi Tiantong, we use the word Shamanism to describe ancient people’s recognition of spirits and gods, and after Juedi Tiantong, we use the word religion 宗教. In a word, “ancient national religion” has been changed to “ancient state religion”(in line 45).
Comments 6: shaman 萨满 = why add 拼音 here? Response: “萨满”has been deleted.(line 92)
Comments 7: "Zhou people 周人's conception" This is not formatted correctly. You don't add the plural marker ('s) to 漢字 like this. This paper is not properly edited. Response: Thank you for pointing out this. The revision has been made as “the Mandate of Heaven of the Zhou”.(line 88-89)
Comments 8: "book of Changes 《周易》:"Again, Book of Changes, not book of Changes. Also the title is 周易, so where the Zhou in the translation? Response: Thanks for pointing out this. “book of Changes” has been modified as “The Zhou Book of Change”(line 98)
Comments 9: "Judging from astronomy to detect when the change, judging from the humanities into the world to" Response: This sentence truly does not make sense and thanks for pointing out this. Revision has been as “we observe the ornamental figures of the sky, and thereby ascertain the changes of the seasons. We look at the ornamental observances of the society, and understand how people are well educated” 觀乎天文,以察時變;觀乎人文,以化成天下.(line98-101)
Comments 10: I'm not reading further. This paper is not coherent or edited properly. I can't make sense of what is being said so far. The author needs to rewrite the whole paper. Response: Indeed, there are lots of language problems in the article which may result in the terrible reading experience. Once again, thank you for taking the time to read. Accordingly, we have thoroughly revised the grammatic errors and modified some expressions in order to make this article apprehensible. However, we can hardly agree with the reviewer’s comment that “this paper is not coherent”. This article is logically organized and the basic outline has been given in the last paragraph of introduction. The article consists of 5 parts and each part has been give a subtitle to show the main argument. We hope the reviewer can kindly continue the reading and we really appreciate the reviewers’ valuable comments. |

Reviewer 2 Report
Comments and Suggestions for Authors
This is a well-researched paper addressing the topic of jue di tian tong 絕地天通 as mentioned in the Shangshu and the Guoyu. Yet, the paper requires substantial revision to be considered for publication. To begin with minor issues, the English translation of some quotes from the Chinese classics simply makes no sense. For instance, it is unclear what the authors mean by “Judging from astronomy to detect when the change, judging from the humanities into the world to” (lines 104-105) or “if you must say 'jue' 绝 is 'he' 合, I would be a bit strange” (line 101). References to the Chinese sources are inconsistent and, sometimes, incorrect. The Shangshu is not “Book of History” (line 150), and the chapter title “Chu yu” 楚語 from the Guoyu certainly cannot be translated as “Chu Dialect”. When quoting from the “Lüxing” 呂刑chapter of the Shangshu the authors use Legge’s English translation yet refer to Kong Yingda (170-191). Legge’s work is not even listed in the bibliography. When translating sentences from other chapters of the Shangshu, the authors appear to provide their own translation, which is, very often, problematic. For instance, “the king of Zhou obeyed the order of Tian and replaced the Shang dynasty to establish the Zhou dynasty” is a very inaccurate rendition of 我有周佑命,将天明威,致王罚,敕殷命,终于帝 (284-285). The respective chapter titles of the Shangshu are never mentioned, just like those of the Lunyu (295, 299). Also, there is a large number of the strangely formulated or incomplete sentences, such as “the natural objects originally worshipped as religion and became the main foundation of Chinese philosophy's "unity of heaven and man"” (lines 633-634). The authors should not submit a work that is filled with so many mistakes and inconsistencies.
The main problem, however, concerns the paper’s contribution to the discussion of jue di tian tong 絕地天通. The authors quote as many as twelve articles dealing with that particular topic and, apart from that, they deal with one of the most fundamental problems of Chinese philosophy, on which a number of influential works have been written. Under such circumstances, the authors should start their article by clarifying in what way their investigation differs from the previous works on the topic and what their paper can contribute to this fundamental discussion.
Comments on the Quality of English LanguageExtensive editing of English language is required.
Author Response
Dear editor
Thank you for giving me the opportunity to submit a revised version of my paper. I really appreciate your time and effort to provide valuable suggestions for the revision of this paper.
Here is a point-by-point response to reviewer's Comments
|
Comments 1: To begin with minor issues, the English translation of some quotes from the Chinese classics simply makes no sense. For instance, it is unclear what the authors mean by “Judging from astronomy to detect when the change, judging from the humanities into the world to” (lines 104-105) or “if you must say 'jue' 绝 is 'he' 合, I would be a bit strange” (line 101). |
|
Response 1:Thank you for pointing this out. We admit that some English translation of Chinese classics is not accurate. Revisions have been made as follows:
1.“Judging from astronomy to detect when the change, judging from the humanities into the world to” is revised as “we observe the ornamental figures of the sky, and thereby ascertain the changes of the seasons. We look at the ornamental observances of the society, and understand how people are well educated”观乎天文,以察时变;观乎人文,以化成天下”(line98-100)
2. “if you must say 'jue' 绝 is 'he' 合, I would be a bit strange” has been modified as “Literally, Juedi Tiantong should be called the division of Tian and man. It was strange to translate 'jue' (isolation絶) into 'he'(unity 合) " (line93-95)
|
|
In addition to these revisions, we have thoroughly checked the whole English translation and made substantial revisions including grammatic errors(line43,line93-95), vocabulary selection(line34-35, line113), sentence restructure(line 38-40, line84-86) and so on(the revised places will not be listed here one by one). We hope the re-submitted manuscript is easier to understand.
Comments 2: References to the Chinese sources are inconsistent and, sometimes, incorrect. The Shangshu is not “Book of History” (line 150), and the chapter title “Chu yu” 楚語 from the Guoyu certainly cannot be translated as “Chu Dialect”. |
|
Response 2: Thanks for pointing this out and we totally agree with the comments. Corresponding revisions have been made as follows: 1. “Book of History”has been revised as Shangshu 尚書(The Discourses of the States)(line142). Lunyu has been revised as The Analects (line 273) 2. “Chu Dialect”has been revised as Chuyu 楚語(the Discourses of the State of Chu)(line143) 3. LYUxing has been revised as Shangshu, Lüxing尚書·呂刑(the Marquis of Lü on Punishments)(line141)
Comments 3:When quoting from the “Lüxing” 呂刑chapter of the Shangshu the authors use Legge’s English translation yet refer to Kong Yingda (170-191). Legge’s work is not even listed in the bibliography. When translating sentences from other chapters of the Shangshu, the authors appear to provide their own translation, which is, very often, problematic. For instance, “the king of Zhou obeyed the order of Tian and replaced the Shang dynasty to establish the Zhou dynasty” is a very inaccurate rendition of 我有周佑命,将天明威,致王罚,敕殷命,终于帝 (284-285). The respective chapter titles of the Shangshu are never mentioned, just like those of the Lunyu (295, 299). Response 3: This is a very helpful comments and we totally agree with it. The accurate English translation of Chinese literature is of great importance. Therefore, we looked up some published English versions of the Chinese classics for reference and have made the following revisions:
1. Firstly, Kong Yingda is the person who gave annotation to the original version written in classical Chinese. Legge made the English translation based on Kong’s work. As the English version of Shangshu translated by Legge is consulted in the article, we have added Legee’s work in the bibliography(line620&621) and deleted Kong Yingda’s book. Secondly, we tried to find a good English translation of each Chinese literature cited in order to ensure the translation accurate. For each published English translation we have consulted in the article, the author has been noted(line 179, line 252, line 259, line 275, line 280), and corresponding references have also been made.(ling 625, line 626, ling 652) However, for some Chinese literature like Zhouyi, we did not find a satisfied English translation, so we looked up some relevant books and give the English translation by ourselves.(line100-103, line573-576)
2.“the king of Zhou obeyed the order of Tian and replaced the Shang dynasty to establish the Zhou dynasty” is a very inaccurate rendition of 我有周佑命,将天明威,致王罚,敕殷命,终于帝 has bee revised as: Duke of Zhou announced (the royal will) to the officers of the Shang dynasty, saying:” The king speaks to this effect:----‘Ye numerous officers who remain from the dynasty of Yin, great ruin came down on Yin from the cessation of forbearance in compassionate Heaven, and we, the lords of Zhou, received its favouring decree. We felt charged with its bright terrors, carried out the punishments which kings inflict, rightly disposed of the appointment of Yin, and finished (the work of ) God (Shangshu,Duoshi. translated by Legge 2013,p.275)尔殷遗多士,弗吊旻天,大降丧于殷。我有周佑命,将天明威,致王罚,敕殷命,终于帝”(尚書·多士)(line252-259)
3. The respective chapter titles of Shangshu尚书 and The Analects论语 have been added. For example, Shangshu, The Numerous Officers尚書·多士(line258); Shangshu, Zhaogao尚書·召誥(line250); The Analects, Yang Huo論語·陽貨(line277)(the revision has been highlighted in the manuscript and will not be listed here one by one)
|
|
Comments 4:Also, there is a large number of the strangely formulated or incomplete sentences, such as “the natural objects originally worshipped as religion and became the main foundation of Chinese philosophy's "unity of heaven and man"” (lines 633-634). The authors should not submit a work that is filled with so many mistakes and inconsistencies. Response3: Thanks for pointing this out, this sentence does not make any sense and has been deleted.
Comments 5: The main problem, however, concerns the paper’s contribution to the discussion of jue di tian tong 絕地天通. The authors quote as many as twelve articles dealing with that particular topic and, apart from that, they deal with one of the most fundamental problems of Chinese philosophy, on which a number of influential works have been written. Under such circumstances, the authors should start their article by clarifying in what way their investigation differs from the previous works on the topic and what their paper can contribute to this fundamental discussion. Response 5: This remark is very important and thank you for pointing this out. The introduction begins with a brief explanation of Juedi Tiantong to give readers basic information on the meaning of this terminology, then the previous academic articles on Juedi Tiantong have been analyzed. So we added a contribution at the end of introduction to show what kind of questions this article aims to solve and how this article contributes in this field.(line114-125)
The explanation of the relationship between Tian and man contained in Juedi Tiantong is conducive to the demonstration of the essence of the unity of heaven and man in ancient China, and will provide a new perspective to understand the relationship between man and nature among different civilizations. Based on the previous research outcomings, this article will conduct the analysis from the perspective of conceptual history and philosophy rather than regarding Juedi Tiantong as a pure historical event, which can better clarify the link between China’s ancient shamanism and Confucianism. This article aims to analyze the impact of Juedi Tiantong on the relationship between Tian and human in ancient China, in order to shed light on how Juedi Tiantong had influenced Chinese ancient people’s understanding of nature, and explore the reason why the term Ziran自然(nature) in Chinese philosophy is different from the term nature in Western modern ecological civilization.
4. Response to Comments on the Quality of English Language |

Round 2
Reviewer 2 Report
Comments and Suggestions for Authors
The Discourses of the States is a usual translation for the Guoyu, not the Shangshu, which is translated as the Book of Documents.
The authors should use either simplified or traditional Chinese characters, but not both forms at the same time.
Comments on the Quality of English Languageok
Author Response
Dear editor
Thank you for giving me the opportunity to submit a revised version of my paper. I really appreciate your time and effort to provide valuable suggestions for the revision of this paper.
|
Point-by-point response to Comments and Suggestions for Authors |
|
Comments 1: The Discourses of the States is a usual translation for the Guoyu, not the Shangshu, which is translated as the Book of Documents. |
|
Response 1: Thanks for pointing this out. “The Book of Documents” has been given in brackets after “Shangshu尚書” to make the translation of the title of these two books more clear. Here is the revised whole sentence:(line143-144)
The term Juedi Tiantong was first mentioned in the chapter Lüxing呂刑(The Marquis of Lü on Punishments) of Shangshu尚書(The Book of Documents), and other literature mentioning this event included Guoyu,Chuyu國語·楚語(The Discourses of the States, the Discourses of the State of Chu)and The Classic of Mountains and Seas山海經.
|
|
Comments 2: The authors should use either simplified or traditional Chinese characters, but not both forms at the same time. |
|
Response 2: Thanks for pointing this out. All the Chinese characters have been changed to traditional ones.(line48,63,142,148,197,259,305,374,520,571) |
